# Inadvertent QRS prolongation by an optimization device-based algorithm in patients with cardiac resynchronization therapy

Kamil Sedláček[1]*, Rostislav Polášek[2], Helena Jansová[3], Domenico Grieco[4], Pavel Kučera[2], Josef Kautzner[3], Darrel P. Francis[5], Dan Wichterle[3,6]

1 1st Department of Internal Medicine–Cardiology and Angiology, University Hospital and Charles University Faculty of Medicine, Hradec Králové, Czech Republic, 2 Cardiology Department, Liberec Regional Hospital, Liberec, Czech Republic, 3 Department of Cardiology, Institute for Clinical and Experimental Medicine, Prague, Czech Republic, 4 Department of Cardiovascular Sciences, Policlinico Casilino of Rome, Rome, Italy, 5 International Centre for Circulatory Health, National Heart and Lung Institute, Imperial College London, Hammersmith Hospital, London, United Kingdom, 6 2nd Department of Internal Cardiovascular Medicine, First Faculty of Medicine, Charles University, Prague, Czech Republic

* kamil.sedlacek@fnhk.cz

**Data Availability Statement:** All relevant data are within the manuscript.

## Abstract

### Background

Device-based algorithms offer the potential for automated optimization of cardiac resynchronization therapy (CRT), but the process for accepting them into clinical use is currently still ad-hoc, rather than based on pre-clinical and clinical testing of specific features of validity. We investigated how the QuickOpt-guided VV delay (VVD) programming performs against the clinical and engineering heuristic of QRS complex shortening by CRT.

### Methods

A prospective, 2-center study enrolled 37 consecutive patients with CRT. QRS complex duration (QRSd) was assessed during intrinsic atrioventricular conduction, synchronous biventricular pacing, and biventricular pacing with QuickOpt-proposed VVD. The measurements were done manually by electronic calipers in signal-averaged and magnified 12-lead QRS complexes.

### Results

Native QRSd was 174 ± 22 ms. Biventricular pacing with empiric AVD and synchronous VVD resulted in QRSd 156 ± 20 ms, a significant narrowing from the baseline QRSd by 17 ± 27 ms, P = 0.0003. In 36 of 37 patients, the QuickOpt algorithm recommended left ventricular preexcitation with VVD of 42 ± 18 ms (median 40 ms; interquartile range 30–55 ms, P <0.00001). QRSd in biventricular pacing with QuickOpt-based VVD was significantly longer compared with synchronous biventricular pacing (168 ± 25 ms vs. 156 ± 20 ms; difference 12 ± 11ms; P <0.00001). This prolongation correlated with the absolute VVD value (R = 0.66, P <0.00001).

**Funding:** The study was supported by the project (Ministry of Health, Czech Republic) for development of research organization 00023001 (IKEM, Prague, Czech Republic) – Institutional support. The funder had no role in study design, data collection and analysis, decision to publish, or preparation of the manuscript.

**Competing interests:** I have read the journal's policy and the authors of this manuscript have the following competing interests: Dr Kautzner reports personal fees from Bayer, Biosense Webster, Boehringer Ingelheim, Daiichi Sankyo, Medtronic, Merck Sharp & Dohme, Merit Medical, and St. Jude Medical (Abbott) for participation in scientific advisory boards, and has received speaker honoraria from Bayer, Biosense Webster, Biotronik, BMS, Boehringer Ingelheim, Daiichi Sankyo, Medtronic, Merck Sharp & Dohme, Mylan, Pfizer, ProMed, and St. Jude Medical (Abbott). All other authors declare no conflicts of interest regarding the paper. This does not alter our adherence to PLOS ONE policies on sharing data and materials.

## Conclusions

QuickOpt algorithm systematically favours a left-preexcitation VVD which translates into a significant prolongation of the QRSd compared to synchronous biventricular pacing. There is no reason to believe that a manipulation that systematically widens QRSd should be considered to optimize physiology. Device-based CRT optimization algorithms should undergo systematic mechanistic pre-clinical evaluation in various scenarios before they are tested in large clinical studies.

## Introduction

Cardiac resynchronization therapy (CRT) improves heart failure symptoms, exercise capacity, morbidity, and mortality in a symptomatic patient with left ventricular (LV) systolic dysfunction and wide QRS complex [1–3]. Optimization of settings of CRT is a complex process because it involves a trade-off balancing of 4 effects: harm from long atrioventricular delay (AVD), harm from short AV delay, harm from excessive LV-first ventriculo-ventricular delay (VVD), and harm from excessive right ventricular (RV)-first VVD. Studies that integrate all 4 and minimise their net effect, by measuring a haemodynamic final common pathway, suggest little benefit from deviating away from a VVD of 0 [4].

We must not assume that moving away from standard configurations is advantageous [5], because if it is possible to improve haemodynamics by changing AV and VV delay, it must also be possible to worsen them. Echocardiography (ECHO) has been used extensively to optimize the programming, but a recent meta-analysis has shown no net benefit on outcomes [6]. Device-based algorithms have the attraction of automatic optimization, perhaps even continuously. However, it is not known if the settings they select are actually better than reference values (such as VVD of 0).

In this study, we examined the behaviour of one of the proprietary algorithms, QuickOpt VV Optimization (St. Jude Medical/Abbott). It was created and implemented to optimize electric resynchronization and minimize the QRS complex duration during the CRT. It has been reported to have similar clinical outcomes as ECHO-based optimization, which in turn are not significantly different from no optimization. Despite the lack of robust evidence supporting its use, and availability of newer promising device-based optimization concepts such as adaptive CRT (aCRT; Medtronic Inc., U.S.A.) and SyncAV (Abbott, U.S.A.), this algorithm is still available in contemporary devices and even used as a comparator in clinical studies.

Testing clinical outcomes is an enormous task, and cannot be carried out on every candidate algorithm, because of the substantial cost. At the conception of this device-based algorithm design the simple heuristic was that CRT is applied in patients with wide QRS complex, with the intention of resynchronizing contraction (manifesting as a narrower QRS complex), and that therefore an optimization process for VVD might be expected to further narrow the QRS complex. We tested whether the Quick-Opt algorithm performs in agreement with this heuristic.

## Methods

Patients undergoing a St. Jude Medical (Abbott) CRT-defibrillator were included in the study conducted in two cardiology centres between June 2013 and May 2015. Local human research ethics committees approved the study protocol and all patients signed informed consent. The indication for CRT was unrelated to the study protocol and was based on the ESC

recommendations for CRT implantation valid at the time of study initiation: patients with persistent heart failure symptoms despite optimal medical therapy, LV ejection fraction $\leq$35%, and QRSd $\geq$120 ms [5]. We excluded patients in whom QuickOpt optimization could not be obtained because of complete AV block or could be biased because of atrial fibrillation with the fast ventricular response. We also excluded patients with the right bundle branch block (RBBB) because they do not represent typical candidates for CRT. The baseline conduction block pattern was classified as a true left bundle branch block (LBBB) when QRS morphology matched Strauss' criteria [7].

The proprietary device-based automatic optimization method QuickOpt available in CRT pacemakers and CRT defibrillators manufactured by St. Jude Medical (Abbott) was tested in this study. The automated QuickOpt optimization algorithm examines intracardiac conduction properties during the spontaneous and ventricular-paced rhythm and calculates the optimal AVD and VVD that should achieve the best electromechanical resynchronization of LV myocardial segments. The specific atrial, RV, and LV sensing and pacing tests performed by the algorithm have been described in detail previously [8]. In brief, the algorithm determines the optimum VVD ($VVD_{opt}$) by the formula: $VVD_{opt} = 0.5 \times (D + \varepsilon)$, where D is the difference between the time of peak intrinsic activation at the LV versus the RV lead and the correction term $\varepsilon$ is the difference in paced interventricular conduction delay (IVCD) for 2 mutually opposite directions of propagation, specifically calculated as IVCD when the LV lead is paced and the delay is sensed at the RV lead minus IVCD when the RV lead is paced and the delay is sensed at the LV lead. During the test, each chamber is paced with a short AVD to prevent fusion of the activation fronts. If the resulting $VVD_{opt}$ is >0 ms, the LV is paced before RV and vice versa. The optimization of the AVD and VVD had been reported to be independent of the lead positions [9].

CRT implantation was performed in the standard way using commercially available CRT devices manufactured by St. Jude Medical (Abbott) using the left subclavian transvenous approach. RV leads were placed in the RV midseptal region whenever feasible. Both bipolar or quadripolar LV leads were positioned in one of the available posterolateral, lateral, or anterolateral tributaries of the coronary sinus. The latency between the QRS complex onset and a local LV electrogram (Q-LV interval) was used to optimize the LV lead position as described previously [10]. If the LV lead electrogram was not recorded within the terminal part of the QRS (specifically, Q-LV/QRSd ratio $\leq$ 0.7 was considered suboptimal), other available veins and lead positions were explored.

The QuickOpt optimization protocol was performed at the end of the implantation procedure using the proprietary St. Jude Medical (Abbott) programmer. Once the automated test had been finished, the consistency of values from repetitive measurements was reviewed before their acceptance. QRSd was assessed during intrinsic AV conduction and biventricular pacing with zero and QuickOpt-proposed VVD from a standard 12-lead ECG (duration of 15 s) obtained by the electrophysiology recording systems: CardioLab (GE Healthcare) or Axiom Sensis XP (Siemens). After completion of the study protocol, study device settings were removed in all patients. The CRT devices were then programmed to an empirical setting according to the preference of the implanting physician, in most instances with the AVD of 150 ms after atrial pacing and 120 ms after atrial sensing, and with the VVD of 0. No routine device optimization was performed.

Purpose-made software was used for data processing. Digitized recordings were exported from the recording system and edited to exclude QRS morphological abnormalities and artifacts. QRS complexes were signal-averaged and magnified. QRSd was manually measured by electronic calipers. This was done by an investigator who was blinded to clinical data and programming mode.

## Statistical analysis

The results are presented as means ± standard deviation or percentages. Pairs of QRSd (or their corresponding differences) were compared by the Student´s t-test for dependent samples. The impact of QuickOpt-based VVD on QRSd was assessed using the Pearson correlation analysis. A P-value <0.05 was considered significant. All analyses were performed using the STATISTICA vers. 12 software (Statsoft, Inc.).

## Results

### Patient characteristics

Thirty-seven patients with CRT devices (defibrillators in 78%) were enrolled. Aetiology of heart failure was ischemic in 57%, the rest of the study participants had non-ischemic cardio-myopathy. Mean ejection fraction was 27 ± 5% and LV end-diastolic diameter 65.2 ± 8.1 mm. Atrial fibrillation was present in 19% of participants. Full details are given in Table 1.

### QRS characteristics without pacing

Native QRSd was 174 ± 22 ms. True LBBB was present in 81% and IVCD in 19% of the study participants. LV lead was implanted with the mean Q-LV interval 136 ± 65 ms, and Q-LV ratio 0.78 ± 0.10 (Table 1).

### QRS characteristics paced with standard settings

In 29/37 patients (78%, P = 0.0005), biventricular pacing with empiric AVD and synchronous VVD settings resulted in a narrower QRSd. The mean paced QRSd with the VVD = 0 was 156 ± 20 ms (median 156 ms, interquartile range 145–165 ms), which was significantly narrower than unpaced native QRS (narrowing of baseline QRSd by 17 ± 27 ms, P = 0.0003, Fig 1).

### QRS characteristics paced with QuickOpt settings

In 36/37 patients (97%, P <0.00001), the QuickOpt algorithm recommended LV preexcitation with VVD of 42 ± 18 ms (median: 40 ms, interquartile range: of 30–55 ms, P <0.00001 vs. zero

**Table 1. Demographic characteristics of the study population (N = 37).**

| | |
|---|---|
| Male sex (%) | 78 |
| Age (years) | 65.6 ± 11.7 |
| Ischemic heart disease (%) | 57 |
| Left-ventricular ejection fraction (%) | 27 ± 5 |
| Left-ventricular end-diastolic diameter (mm) | 65.2 ± 8.1 |
| Atrial fibrillation (%) | 19 |
| Implantable-cardioverter defibrillator (%) | 78 |
| True left-bundle branch block (%) | 81 |
| IVCD (%) | 19 |
| Native QRSd (ms) | 174 ± 22 |
| Q-LV (ms) | 136 ± 65 |
| Q-LV / QRSd ratio | 0.78 ± 0.10 |

Values are means ± standard deviation or percentages.

IVCD, non-specific intraventricular conduction delay; Q-LV, the interval between QRS onset and local LV lead electrogram during the intrinsic rhythm.

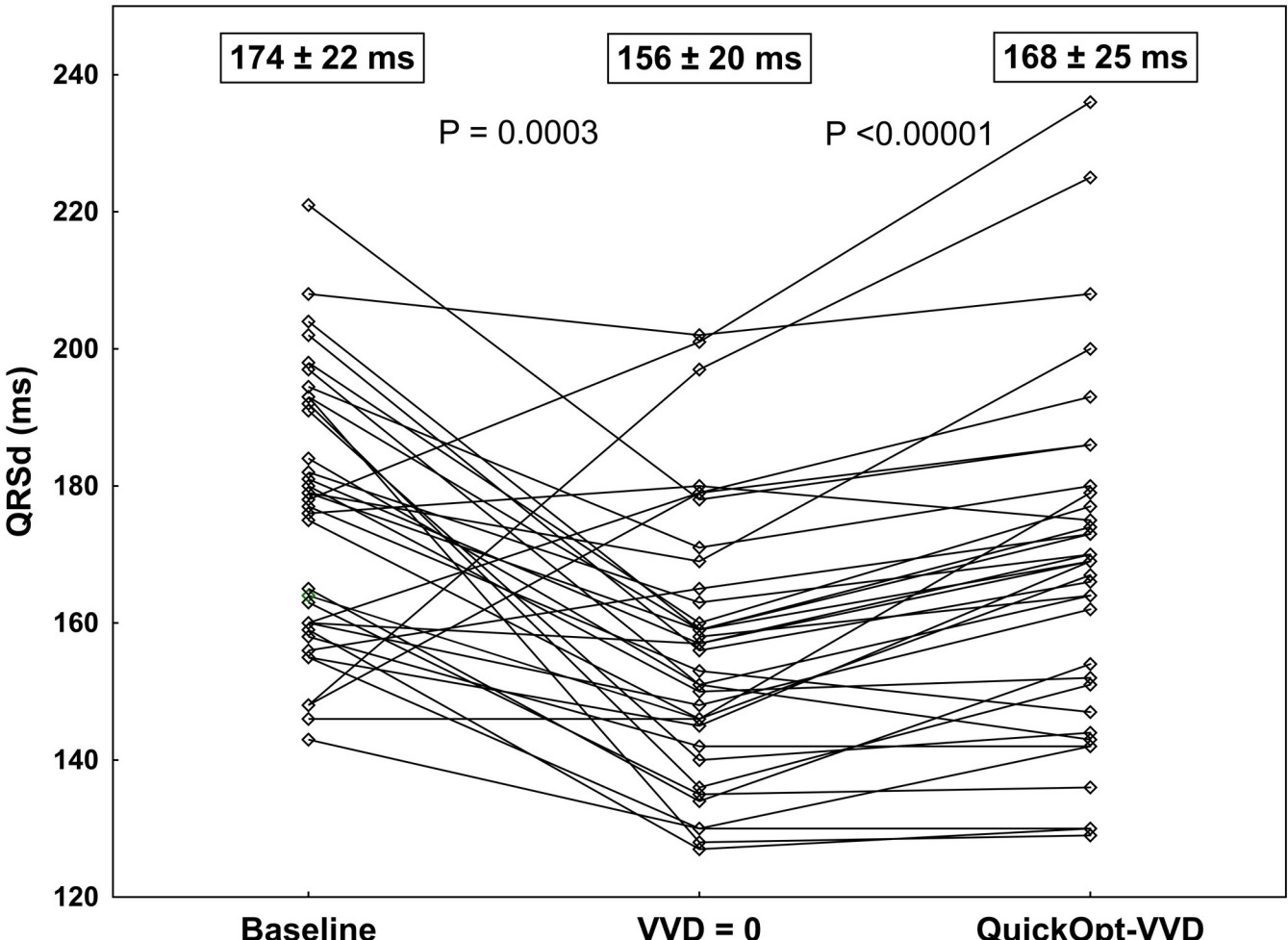

**Fig 1. QRSd with synchronous biventricular pacing and after QuickOpt optimization.** QRSd, QRS duration; VVD, ventriculo-ventricular delay.

VVD, Fig 2). The resulting QRSd ranged from 127 to 202 ms (mean 168 ± 25 ms). This was significantly wider than paced QRS with standard synchronous settings (QRSd difference 12 ± 11 ms, P <0.00001; Fig 3) but not significantly narrower than native unpaced QRS (mean narrowing vs. baseline QRSd by 6 ± 32 ms, P = 0.26, Fig 1).

### Impact of VVD on QRS widening

The relative QRSd prolongation due to the QuickOpt programming correlated positively with the absolute value of VVD (R = 0.66, P <0.00001, Fig 4).

### Discussion

This study demonstrates that the QuickOpt automated device-based algorithm compares poorly with synchronous biventricular pacing and results in widening of the QRS complex through systematic and substantial LV preexcitation.

The role of the QRS narrowing with CRT has been subject to controversy although implanters have been using it extensively as a readily available and intuitive measure of adequate

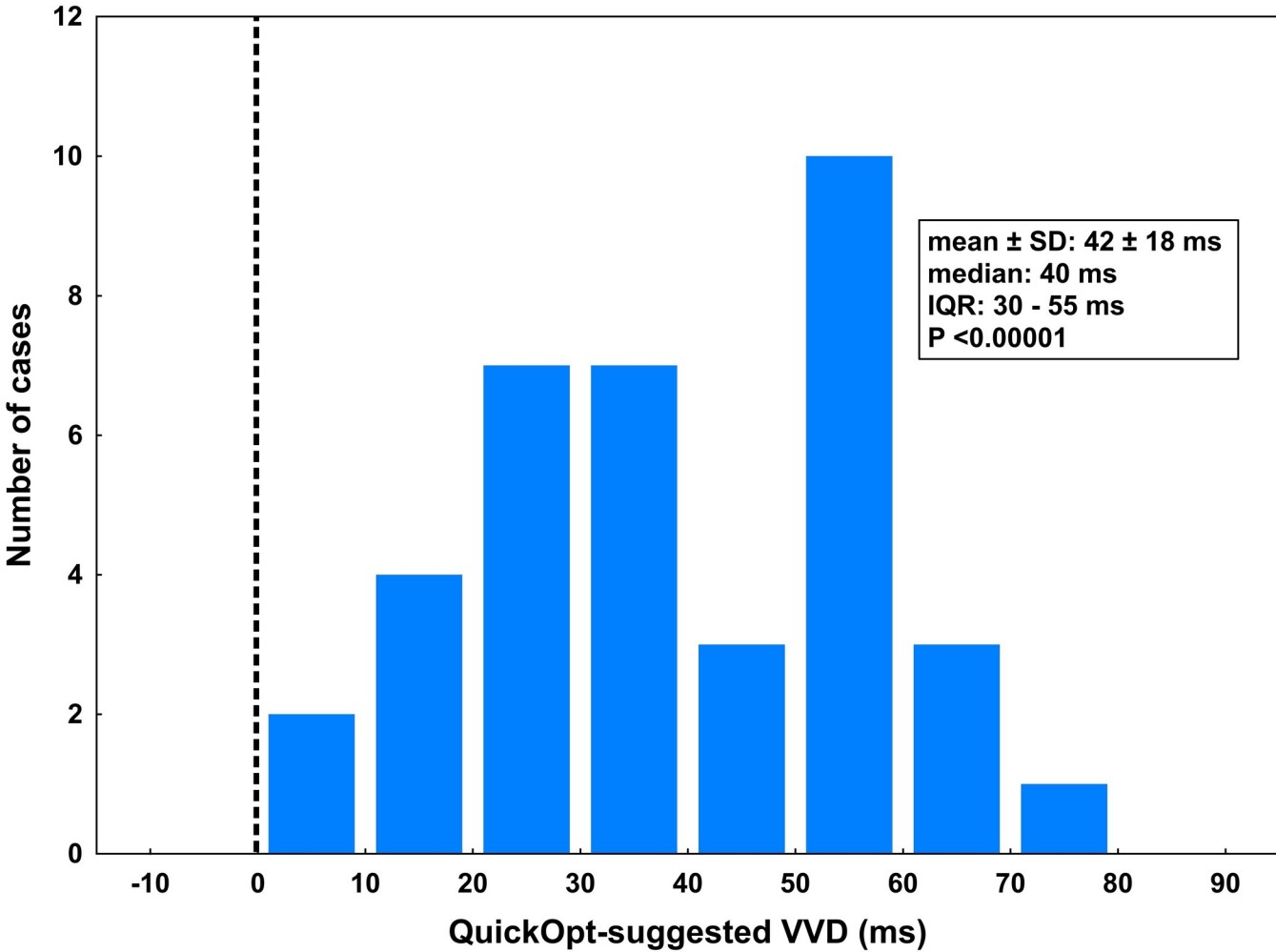

**Fig 2. Distribution of VVD as suggested by the QuickOpt algorithm.** Left ventricular preexcitation was suggested in all but one patient with VVD = 0. IQR, interquartile range; VVD, ventriculo-ventricular delay.

electric resynchronization. Recent studies in patients with LBBB have confirmed that QRSd indeed serves as a robust biomarker and endpoint of the CRT implant [11, 12] and that post-operative QRS prolongation is associated with increased mortality risk during follow-up [13].

The prolongation of the QRS complex resulting from the QuickOpt use is an unexpected and untoward result of a device-based optimization algorithm. It may potentially have harmful effects, especially in patients with true LBBB and optimal LV lead position in whom Q-LV and LV-RV delay are among the longest and, consequently, higher LV preexcitation with more prominent adverse impact on QRSd is suggested by QuickOpt.

CRT optimization is a complex process aiming at the maximization of the therapeutic benefit. Optimal outcomes of CRT are related to multiple factors in the preimplantation phase (selection of candidates), procedural phase (quality of device implantation), and post-implant phase (appropriate device programming, troubleshooting, and careful clinical follow-up). The acute hemodynamic benefit of AVD optimization was known before the advent of CRT in the experimental era of dual-chamber pacing for heart failure [14]. After the introduction of CRT, VVD optimization was studied to optimize the outcomes of CRT patients. The hemodynamic

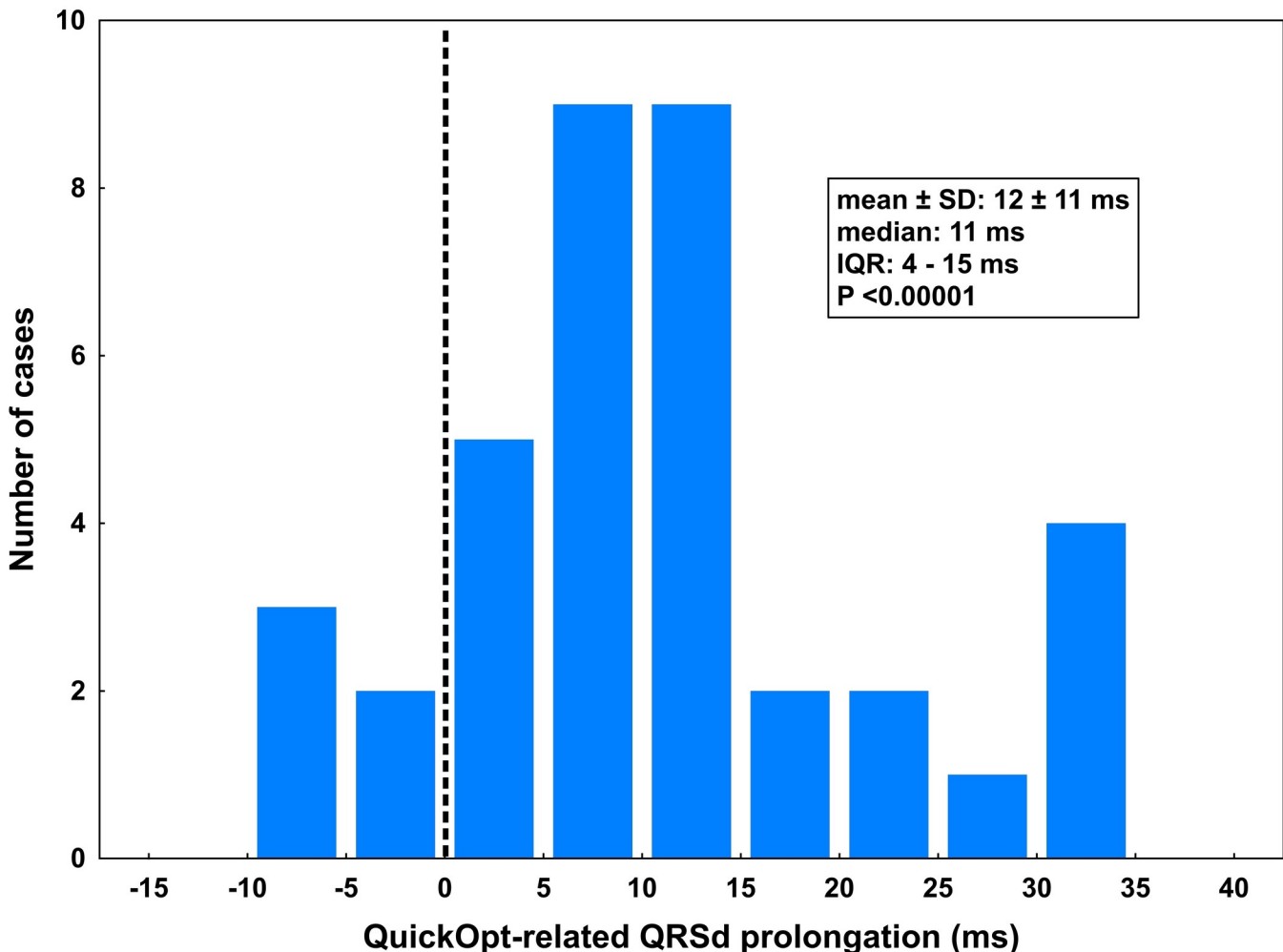

**Fig 3. Distribution of relative QRSd prolongation due to QuickOpt optimization.** Relative QRSd prolongation is the difference in QRSd between the QuickOpt-guided and synchronous biventricular pacing. QRSd was prolonged in all but five patients: not changed in 2 patients; shortened by 5–8 ms in 3 patients. IQR, interquartile range; QRSd, QRS duration.

assessment was used frequently in the early phase of the CRT era, but ECHO soon became a standard non-invasive tool guiding the optimization of CRT devices [15, 16]. However, in clinical practice, routine optimization is performed infrequently and more importantly, ECHO failed to demonstrate the reproducible benefit of both AVD and VVD optimization when tested in clinical trials [6]. Therefore, practice guidelines do not support routine individual AVD and VVD optimization using ECHO. Although ECHO is not a meaningful comparator or gold standard for comparison with other methods, it has been used in various scenarios, including validation of device-based algorithms.

Device-based optimization algorithms have been introduced with the hope for unbiased, repeatable, or even continuous automated device optimization. The QuickOpt algorithm has been implemented as an automated programming optimization tool in St. Jude Medical (now Abbott) implantable devices for more than a decade. It was from the outset designed to optimize electrical resynchronization and minimize resulting QRS complex duration on biventricular pacing. The QuickOpt formula is known, but its rationale and pathophysiological

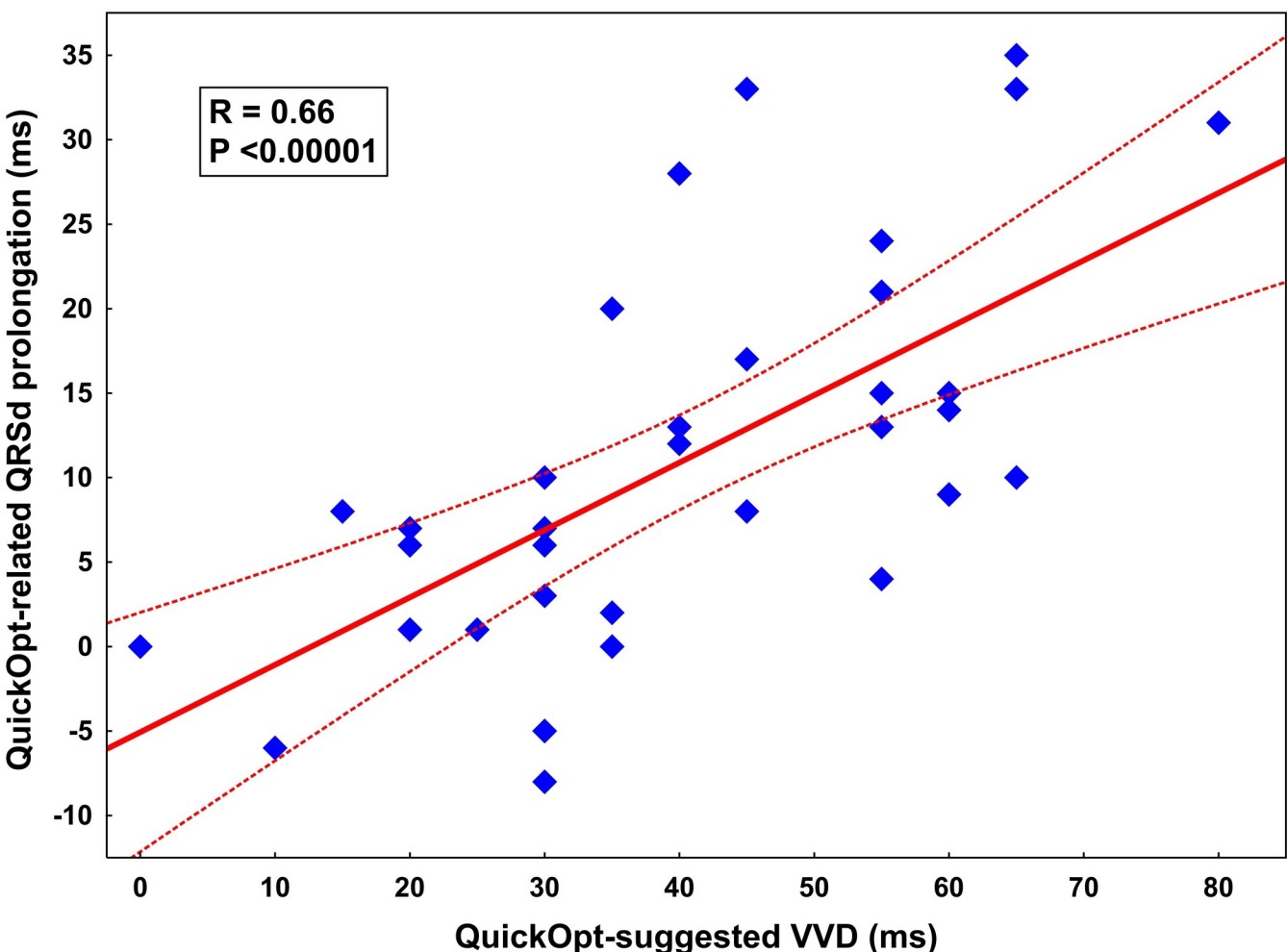

**Fig 4. Correlation between QuickOpt-suggested VVD and QuickOpt-related QRSd prolongation.** QRSd, QRS duration; VVD, ventriculo-ventricular delay.

background were never disclosed, and the algorithm was not preclinically tested. By its apparently irrational computation algorithm, it systematically suggests LV preexcitation in LBBB and IVCD and RV preexcitation in RBBB. In real life CRT, unless there is a particular reason for the stimulus-to-QRS delay (such as in LV pacing from areas of slow conduction or scar), an empiric zero VVD has been shown to perform better than alternatives most of the time [5, 6]. The RV to LV sensed interval, which is used as the defining value of the QuickOpt formula, is an intracardiac approximation of the Q-LV interval. The Q-LV interval reflects the presence of LBBB or IVCD and helps to identify late activated areas in the left ventricle. There is however no physiological rationale for its use to predict adequate VVD, unless it reflects conduction latency in scar tissue. This can only be quantified by the measurement of the stimulus-to-QRS interval, ideally by 12-lead ECG. Such a measurement cannot be performed by the device algorithm itself. For the QuickOpt algorithm, the RV to LV interval would only be useful in case the resulting CRT setting would utilize fusion on native conduction, but it is unlikely that the current formula would be adequate.

In some studies that compared VVD optimization by QuickOpt and ECHO, optimized aortic velocity-time integrals (VTI) instead of corresponding VVDs were analysed and highly

significant correlations were misleadingly considered proof of the agreement between both methods [9, 17]. Unfortunately, this is conceptually wrong statistics. Even tight correlation provides little information in this respect because it simply reflects much higher interindividual variability of VTI compared to variability due to VVD programming.

In contrast, there was a poor agreement in optimal VVD determined by alternative optimization methods and QuickOpt in other studies. QuickOpt VVD correlated neither with VVD at a maximum of invasively measured LV dp/dt [18] nor with VVD at a maximum of aortic VTI [19, 20]. In these studies, there was no benefit from QuickOpt VVD optimization compared to synchronous biventricular pacing [18] and compared to ECHO-based optimization [19, 20].

Several clinical studies failed to document clinical benefit from QuickOpt optimization. The largest of them was the FREEDOM trial that included 1647 patients and compared the QuickOpt AVD and VVD optimization with the standard clinical practice. After 12 months of follow-up, it did not show any improvement in the clinical heart failure composite score [21]. Its in-depth analysis concerning potential harm by the use of QuickOpt algorithm cannot be provided, since the study results have never been published *in extenso*.

Two recent studies used the QuickOpt algorithm as an additional comparator to conventional CRT programming and/or novel SyncAv algorithm (Abbott, U.S.A.). Wang et al. used the QuickOpt algorithm along with two modes of the Sync AV algorithms (default 50 ms offset vs. optimized offset minimizing QRS complex duration) [22]. The native QRS complex duration was reduced by the QuickOpt algorithm by approximately 20 ms but no details were given on its specific performance in comparison with an empiric CRT with a VVD = 0. Similarly, in a study by AlTurki et al., both QuickOpt and AV Sync programming were used to modify the original empirical clinician-preferred programming [23]. In this study, the Quick-Opt programming prolonged the QRS complex of the empirical CRT by 8 ms whereas the Syn-cAV algorithm was associated with the mean QRS complex reduction by 14 ms when compared with the original empirical CRT setting. None of these studies focused on the QuickOpt algorithm performance in detail and none of the two compared native, empirical, and Quick-Opt QRS durations as the primary outcome.

What our study also adds is that the better the electric position of the LV lead according to our current knowledge (specifically, the later electrical position of the LV lead), the longer VVD the QuickOpt algorithm suggests that results in theoretically optimized but in reality prolonged and likely electrically desynchronized QRS complex.

Device-based CRT optimization algorithms should be validated and tested in the same way as any other therapeutic modalities used in clinical practice. A detailed description of proposed algorithms should be followed by methodologically sound mechanistic studies of their performance in various patient and procedural scenarios. Only after this phase and after oversight by regulatory authorities, such algorithms should be approved for testing in clinical studies comparing them with other robust therapeutic standards and for trials with surrogate and clinical outcome endpoints.

## Limitations

Although the study is small, it should not be considered a true limitation as a highly significant result indicates the magnitude of the problem. The QRSd was the only objective of the study. Although systematic prolongation of the QRSd associated with QuickOpt programming seems to be a worrisome and undesired result of optimization, the study was not designed to investigate the potential adverse impact on other outcome measures, like acute haemodynamics or even clinical endpoints. On the other hand, the QRS width intuitively reflects the electric

resynchronization and the QRS narrowing is associated with improved outcomes in CRT recipients. In this study, we did not take into account the effect of the intrinsic AV interval and the CRT systems within this study had an empiric setting according to the implanting physician preference and in line with current guidelines. On one hand, AV delay optimization may enhance fusion with the native conduction, on the other hand, the empirical setting (AVD 120–150 ms) targeting at maximizing biventricular pacing was likely beneficial in reducing the heterogeneity of the outcome measure under study.

## Conclusion

The automated QuickOpt optimization algorithm, due to apparent misconception in its computation, results in excessive VVDs, which translate in systematic prolongation of the QRSd compared to currently recommended synchronous biventricular pacing. There is no reason to believe that a manipulation that systematically widens QRSd should be considered to optimize physiology. At best, this effect should be assumed to be neutral; at worst, it may be harmful. Given its untoward behaviour, lack of supportive clinical studies, and the availability of newer algorithms supported by clinical studies, we believe the QuickOpt algorithm should be abandoned from clinical and trial use and removed from contemporary devices. Newly introduced algorithms should be subjected to detailed pre-clinical scrutiny in various individual scenarios (e.g., in different conduction patterns) before being tested in large clinical studies.

## Author Contributions

**Conceptualization:** Kamil Sedláček, Rostislav Polášek, Josef Kautzner, Dan Wichterle.

**Data curation:** Kamil Sedláček, Rostislav Polášek, Helena Jansová, Pavel Kučera, Dan Wichterle.

**Formal analysis:** Kamil Sedláček, Dan Wichterle.

**Investigation:** Domenico Grieco, Pavel Kučera.

**Methodology:** Kamil Sedláček, Domenico Grieco, Josef Kautzner.

**Project administration:** Kamil Sedláček, Josef Kautzner.

**Resources:** Josef Kautzner.

**Supervision:** Rostislav Polášek, Josef Kautzner, Dan Wichterle.

**Visualization:** Dan Wichterle.

**Writing – original draft:** Kamil Sedláček, Josef Kautzner, Dan Wichterle.

**Writing – review & editing:** Kamil Sedláček, Darrel P. Francis, Dan Wichterle.

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
