## [Decision Letter · Decision Letter 0]

7 Dec 2021

PONE-D-21-32458Inadvertent QRS prolongation by an optimization device-based algorithm in patients with cardiac resynchronization therapyPLOS ONE

Dear Dr. Sedláček,

Thank you for submitting your manuscript to PLOS ONE. After careful consideration, we feel that it has merit but does not fully meet PLOS ONE’s publication criteria as it currently stands. Therefore, we invite you to submit a revised version of the manuscript that addresses the points raised during the review process.

We look forward to receiving your revised manuscript.

Kind regards,

Moshe Swissa, MD

Academic Editor

PLOS ONE

Journal Requirements:

"The study was supported by the project (Ministry of Health, Czech Republic) for development of research organization 00023001 (IKEM, Prague, Czech Republic) – Institutional support"

"I have read the journal's policy and the authors of this manuscript have the following competing interests: Dr Kautzner reports personal fees from Bayer, Biosense Webster, Boehringer Ingelheim, Daiichi Sankyo, Medtronic, Merck Sharp & Dohme, Merit Medical, and St. Jude Medical (Abbott) for participation in scientific advisory boards, and has received speaker honoraria from Bayer, Biosense Webster, Biotronik, BMS, Boehringer Ingelheim, Daiichi Sankyo, Medtronic, Merck Sharp & Dohme, Mylan, Pfizer, ProMed, and St. Jude Medical (Abbott).

All other authors declare no conflicts of interest regarding the paper."

Additional Editor Comments:

According to the comments of the reviewers it can be seen that in the two main foci of the manuscript the method and the statistics sections an profound changes are needed.

Reviewers' comments:

Reviewer's Responses to Questions

**Comments to the Author**

1. Is the manuscript technically sound, and do the data support the conclusions?

Reviewer #1: Partly

Reviewer #2: Yes

2. Has the statistical analysis been performed appropriately and rigorously? 

Reviewer #1: No

Reviewer #2: Yes

3. Have the authors made all data underlying the findings in their manuscript fully available?

Reviewer #1: No

Reviewer #2: Yes

4. Is the manuscript presented in an intelligible fashion and written in standard English?

Reviewer #1: Yes

Reviewer #2: Yes

5. Review Comments to the Author

Reviewer #1: PONE-D-21-32458: Statistical review

SUMMARY: This is a study of the performance of a device-driven algorithm for ventriculo-ventricular delay against QRS complex shortening by cardiac resynchronization therapy. The statistical analysis relies on student tests for dependent data. I have two major concerns about this paper.

MAJOR ISSUES

1. Although repeated measures ANOVA would be the natural approch here, dependent-data pairwise t-test comparisons are acceptable. However, both t-tests and ANOVA methods require normality of the data. Either the authors are capable to provide evidence of normality of the measurements by a formal test or they should repeat the analysis by nonparametric methods that don't require the normality assumption.

2. The statistical analysis assumes a homogeneous sample. However, Table 1 displays heterogeneous characteristics of the observed subjects. Again, I see two options here. Either the authors argue that these characteristics are not confounding factors. Or, pairwise comparisons should be adjusted for these characteristics, e.g. using a linear mixed model for repeated measures, where the information of Table 1 is included as a battery of covariates.

Reviewer #2: The concept of evaluating the real effect of automated QRS optimization (QuickOpt in this case)is very interesting. The demonstrated QRS prolongation after optimization is important finding.

It should be noted that while shortening of the QRS was associated with improved outcomes in some studies, the reasons for wide QRS could be different and not always related to conduction system dysfunction. Also, pacing the site with latest intrinsic activation may not necessarily lead to better outcomes.

In order to better evaluate the article the following should be addressed:

1. Do we have any clinical outcomes?

2. Do we have data stratified by ischemic versus nonischemic CMP?

3. Do we have data stratified by the LV lead position?

4. How QRS optimization affected patients with initially more narrow versus more wide QRS, same about narrow versus wide VDD=0?

5. |Did optimization improve those patients, who had wider VVD=0 than intrinsic QRS (I counted at least 7) and who they were (type of conduction, etiology, lead position, etc)?

6. Consider recalculate statistics for LBBB only (without IVCD) patients.

6. PLOS authors have the option to publish the peer review history of their article (what does this mean?). If published, this will include your full peer review and any attached files.

Reviewer #1: No

Reviewer #2: No

---

## [Author Response · Author response to Decision Letter 0]

12 Dec 2021

All requested amendments to the manuscript have been done and are described in the Cover Letter and in the Response to Reviewers

---

## [Decision Letter · Decision Letter 1]

8 Aug 2022

PONE-D-21-32458R1Inadvertent QRS prolongation by an optimization device-based algorithm in patients with cardiac resynchronization therapyPLOS ONE

Dear Dr. Sedláček,

Thank you for submitting your manuscript to PLOS ONE. After careful consideration, we feel that it has merit but does not fully meet PLOS ONE’s publication criteria as it currently stands. Therefore, we invite you to submit a revised version of the manuscript that addresses the points raised during the review process.

We look forward to receiving your revised manuscript.

Kind regards,

Lucinda Shen, MSc

Staff Editor

on behalf of 

Moshe Swissa, MD

Academic Editor

PLOS ONE

Journal Requirements:

Additional Editor Comments (if provided):

Reviewers' comments:

Reviewer's Responses to Questions

**Comments to the Author**

1. If the authors have adequately addressed your comments raised in a previous round of review and you feel that this manuscript is now acceptable for publication, you may indicate that here to bypass the “Comments to the Author” section, enter your conflict of interest statement in the “Confidential to Editor” section, and submit your "Accept" recommendation.

Reviewer #1: All comments have been addressed

Reviewer #3: (No Response)

Reviewer #4: All comments have been addressed

2. Is the manuscript technically sound, and do the data support the conclusions?

Reviewer #1: (No Response)

Reviewer #3: Yes

Reviewer #4: Yes

3. Has the statistical analysis been performed appropriately and rigorously? 

Reviewer #1: (No Response)

Reviewer #3: Yes

Reviewer #4: Yes

4. Have the authors made all data underlying the findings in their manuscript fully available?

Reviewer #1: (No Response)

Reviewer #3: Yes

Reviewer #4: Yes

5. Is the manuscript presented in an intelligible fashion and written in standard English?

Reviewer #1: (No Response)

Reviewer #3: Yes

Reviewer #4: Yes

6. Review Comments to the Author

Reviewer #1: (No Response)

Reviewer #3: The authors tested the QuickOpt algorithm still present in St Jude/Abbot CRT devices for electrical optimization of the VV interval programming, and found that use of this algorithm, in patients with LBBB caused a lengthening of the paced QRS compared to empirical programming with simultaneous LV-RV pacing. The results are relevant. The large Freedom trial, funded by St Jude, finished recruitment in 2016 and has yet to be published- although the online results in the clinical trials database indicate that the trial was negative, and raises the suspicion that the results deliberately remain unpublished. The authors perform a service by highlighting that the use of QuickOpt remains untested, and insinuate that it also may be harmful.

The authors have revised the statistical analysis and adequately addressed a previous round of revisions.

Reviewer #4: Evaluation of resynchronization optimization through manual versus automatic method based on device algorithms is a very interesting idea, since there is vast literature that evidences clinical benefits in the narrowing of QRS complex.

CONSIDERATIONS

1-Do we have clinical outcomes?

2- Is there a relationship between the degree of QRS reduction and coronary sinus electrode position?

3-Is there a difference in the degree of narrowing QRS in regarding to gender?

4-Do we have echocardiogram outcomes and degree of narrowing QRS complex?

7. PLOS authors have the option to publish the peer review history of their article (what does this mean?). If published, this will include your full peer review and any attached files.

Reviewer #1: No

Reviewer #3: No

Reviewer #4: **Yes: **RAFAEL DIAMANTE

---

## [Author Response · Author response to Decision Letter 1]

10 Aug 2022

Response to reviewers has been provided in an attached file as required.

---

## [Editor Report · Decision Letter 2]

13 Sep 2022

Inadvertent QRS prolongation by an optimization device-based algorithm in patients with cardiac resynchronization therapy

PONE-D-21-32458R2

Dear Dr. Sedláček,

We’re pleased to inform you that your manuscript has been judged scientifically suitable for publication and will be formally accepted for publication once it meets all outstanding technical requirements.

Kind regards,

Moshe Swissa, MD

Academic Editor

PLOS ONE

Additional Editor Comments (optional):

The authors have revised the statistical analysis and adequately addressed a previous round of revisions.
---

## [Editor Report · Acceptance letter]

15 Sep 2022

PONE-D-21-32458R2 

Inadvertent QRS prolongation by an optimization device-based algorithm in patients with cardiac resynchronization therapy 

Dear Dr. Sedláček:

I'm pleased to inform you that your manuscript has been deemed suitable for publication in PLOS ONE. Congratulations! Your manuscript is now with our production department. 

Kind regards, 

on behalf of

Prof Moshe Swissa 

Academic Editor

PLOS ONE